# Characterizing Intraindividual Podocyte Morphology In Vitro with Different Innovative Microscopic and Spectroscopic Techniques

**DOI:** 10.3390/cells12091245

**Published:** 2023-04-25

**Authors:** Annalena Kraus, Victoria Rose, René Krüger, George Sarau, Lasse Kling, Mario Schiffer, Silke Christiansen, Janina Müller-Deile

**Affiliations:** 1Institute for Nanotechnology and Correlative Microscopy, INAM, 91301 Forchheim, Germany; 2Department of Nephrology and Hypertension, Universitätsklinikum Erlangen, Friedrich-Alexander-University (FAU) Erlangen-Nürnberg, 91054 Erlangen, Germany; 3Fraunhofer Institute for Ceramic Technologies and Systems IKTS, 91301 Forchheim, Germany; 4Leuchs Emeritus Group, Max Planck Institute for the Science of Light, 91058 Erlangen, Germany; 5Research Center on Rare Kidney Diseases (RECORD), Universitätsklinikum Erlangen, 91054 Erlangen, Germany; 6Physics Department, Freie Universität Berlin, 14195 Berlin, Germany

**Keywords:** single-cell RNA sequencing, podocytes, foot processes, filopodia, light microscopy, Raman spectroscopy, scanning electron microscopy, atomic force microscopy, scanning ion-conductance microscopy, nanoGPS tracking

## Abstract

Podocytes are critical components of the glomerular filtration barrier, sitting on the outside of the glomerular basement membrane. Primary and secondary foot processes are characteristic for podocytes, but cell processes that develop in culture were not studied much in the past. Moreover, protocols for diverse visualization methods mostly can only be used for one technique, due to differences in fixation, drying and handling. However, we detected by single-cell RNA sequencing (scRNAseq) analysis that cells reveal high variability in genes involved in cell type-specific morphology, even within one cell culture dish, highlighting the need for a compatible protocol that allows measuring the same cell with different methods. Here, we developed a new serial and correlative approach by using a combination of a wide variety of microscopic and spectroscopic techniques in the same cell for a better understanding of podocyte morphology. In detail, the protocol allowed for the sequential analysis of identical cells with light microscopy (LM), Raman spectroscopy, scanning electron microscopy (SEM) and atomic force microscopy (AFM). Skipping the fixation and drying process, the protocol was also compatible with scanning ion-conductance microscopy (SICM), allowing the determination of podocyte surface topography of nanometer-range in living cells. With the help of nanoGPS Oxyo^®^, tracking concordant regions of interest of untreated podocytes and podocytes stressed with TGF-β were analyzed with LM, SEM, Raman spectroscopy, AFM and SICM, and revealed significant morphological alterations, including retraction of podocyte process, changes in cell surface morphology and loss of cell-cell contacts, as well as variations in lipid and protein content in TGF-β treated cells. The combination of these consecutive techniques on the same cells provides a comprehensive understanding of podocyte morphology. Additionally, the results can also be used to train automated intelligence networks to predict various outcomes related to podocyte injury in the future.

## 1. Introduction

Podocytes are specialized cells of the glomerular filtration barrier whose function is highly dependent on intact and highly specialized morphology. Podocytopathies are kidney diseases, in which direct or indirect podocyte injury drives proteinuria, or even nephrotic syndrome [1]. Primary and secondary foot processes of podocytes are highly dependent on an intact actin cytoskeleton [2,3,4].

In order to fulfil its functions, the cytoskeleton must form highly organized dynamic structures, such as stress fibers, lamellipodia (flattened extensions) and filopodia (long, slender, tapering extensions) [3]. When extracellular environment changes, these structures are disassembled and remodeled to meet new requirements [5]. Precise organization and regulation of the actin cytoskeleton in podocytes are essential for maintenance of its normal structure and function. However, not much is known about the morphology of lamellipodia and filopodia, especially in cultured podocytes, as the complex morphological changes and chemical alterations that occur during podocyte injury are challenging to capture by conventional microscopes.

Despite its clinical significance, the lack of ground truth data for the morphological and chemical characterization of these special podocyte structures presents a major challenge in understanding the underlying mechanisms of podocyte injury, and in developing effective therapeutic strategies. To gain a comprehensive understanding of the morphological and chemical changes in podocytes during injury, it is crucial to use a combination of complementary microscopic and spectroscopic techniques. Scanning electron microscopy (SEM) is a widely used method for conducting such studies, due to high-resolution imaging on nanometer range. Still, it requires multiple complex fixation procedures, making it impossible to analyze a living sample. Therefore, there is a critical need to develop new tools to study alterations of podocyte morphology in living samples. Scanning ion conductance microscopy (SICM) is a technique that enables high-resolution, non-optical 3D-imaging of living cell surfaces with complex morphology. So far, SICM was only performed on whole glomeruli [6,7], but not yet on cultured human podocytes.

As podocytes are terminally differentiated cells, most researchers use conditionally immortalized podocyte (ciPodocytes) lines carrying a temperature-dependent SV40 antigen for cell culture experiments. These immortalized podocytes proliferate at a permissive temperature of 33 °C, and start terminal differentiation when cultured at 37 °C [8]. The development of immortalized podocyte lines was thought to provide a tool to study podocyte biology and behavior. Even though immortalized podocytes develop foot process-like structures, they are usually more rudimentary compared to the in vivo phenotype. However, there is barely any knowledge about foot processes and other cell protrusions of cultured podocytes. Time-lapse micrographs of cultured podocytes revealed that podocyte cell protrusions consisted of “spikes”, resulting from lamellipodia that has extended, attached to the substrate or to adjacent cells, and finally formed spike-like processes as a result of subsequent retraction [9]. However, podocyte cell surface, foot processes and filopodia have not been visualized within the same cell with different methods before. Still, this aspect is urgently needed, as dramatic differences in marker expression, response to toxins, and motility between podocytes, and even between similarly derived lines were identified [9]. Moreover, scRNASeq revealed that individual cells can behave extremely different [9,10,11]. Although heterogeneity between single cells is obvious in tissues, it can also be observed in cells in vitro, cultured under identical conditions [12]. Variation in cells of the same population can regard gene expression, DNA sequence, metabolomic properties and proteome, but also cell size and morphology, due to differences in function, changes in metabolism, or simply because of being in different phases of the cell cycle [11,12,13,14,15].

Conclusions derived from cell culture experiments most often rely on a combination of different techniques to analyze cell morphology, marker expression or behavior. Usually, different cells, or a population of various cells, are used, due to differences in pre- and intra-analytical needs. However, even if cells that are adjacent to each other do not give the same outcome, the results of multiple techniques performed in different cells may not build on each other, leading to misinterpretation or confusing data.

Here, we present data from scRNASeq in ciPodocytes, confirming that marker genes involved in cell morphology and cell process formation are highly variable between single cells of the same experiment.

Therefore, we aimed to develop a correlative microscopy workflow, allowing to contextualize podocytes in high-resolution, in a non-invasive and label-free manner, and, most importantly, allowing to measure the exact same cells with different methods. We established a preanalytical workflow concerning specimen holder size and material, substrate coating, drying and fixation, that worked best for all methods, and defined the best sequence for the different measurements, in order to not interfere with the next technique. A workflow protocol was developed that combines LM, Raman spectroscopy, SEM and AFM one after another, not only in the same sample, but also in the same cells, with the help of nanoGPS technology [15]. LM was used to obtain an overview of the entire coverslip, populated with podocytes, which then was used to specifically define the regions of interest (ROI). Raman spectroscopy was performed at the same position to analyze the chemical composition of the cells. SEM was subsequently used to visualize the ultrastructure of the podocytes, including foot processes and filopodia, at high-resolution. Finally, AFM could confirm electron microscopy measurements, and, additionally, imaged the topography of the cell. The protocol was also compatible with measuring living cells when skipping fixation and drying, allowing, for the first time, to perform scanning ion conductance microscopy (SICM) in cultured human podocytes. We used untreated and TGF-β-stimulated (simulate cell stress [16]) ciPodocytes for each imaging method, and focused on differences in podocyte morphology and surface structures. All these methods, correlatively combined, might lead to new insights into podocyte morphology.

## 2. Materials and Methods

### 2.1. Cell Culture

Human ciPodocytes (kindly gifted from Moin Saleem; Children’s and Renal Unit and Bristol Renal, University of Bristol) were proliferated under permissive conditions at 33 °C. Undifferentiated podocytes were dissociated using 1× trypsin-EDTA for 5 min at 33 °C, and 3.5 × 10^3^ cells per cm^2^ were seeded onto uncoated, or chromium-, platinum- and laminin 511 (RLS 115 biogenes; 2.5 µg/mL)-coated glass slides, or plastic cover slides. When cultivated at 37 °C, the SV40 large T-antigen was inactivated for terminal cell differentiation. Podocytes were differentiated on glass cover slides in RPMI 1640 Medium (Roth, Karlsruhe, Germany), supplemented with 10% fetal calf serum, 1% penicillin/streptomycin and 0.1% insulin. After 7 days at 37 °C, cells were starved with 1% fetal calf serum overnight, and either left untreated for another 48 h, or treated with 5 ng/mL TGF-β for 48 h, to induce cell damage before fixation and imaging.

### 2.2. scRNASeq

Human ciPodocytes, differentiated for 7 days at 37 °C, were dissociated with a gentle cell dissociation reagent (e.g., accutase). After washing once with PBS 3 × 10^5^, cells were resuspended in 500 µL PBS, containing 2% FCS, for scRNAseq analysis. scRNASeq was performed at Core Unit Next Generation Sequencing of Friedrich-Alexander University, Erlangen-Nürnberg, using a droplet-based method. Libraries were prepared using the Chromium controller (10× Genomics, Pleasanton, CA, USA), in conjunction with the single-cell 3′ v2 kit. Briefly, the cell suspensions were diluted in nuclease-free water, according to manufacturer instructions. cDNA synthesis, barcoding, and library preparation were then carried out, according to the manufacturers’ instructions. Using default parameters, we obtained the unfiltered feature-barcode matrix per sample by passing the demultiplexed FASTQs to Cell Ranger v6.0.0 ‘count’ command. The reads were aligned against the genome reference from the 10× Genomics pre-built human genome reference (GRCh38; GENCODE version 37). Seurat v4.3.0 (Developed by Paul Hoffman, Satija Lab and Collaborators; New York, NY, USA; [17]) was used for all subsequent analyses. The Seurat object was constructed using the unfiltered feature-barcode matrix. We used various quality filters to remove those cells into any of these categories: Too few total transcripts (<300), possible debris with too few genes expressed (<320), possibly stressed or damaged or cellular stress, and apoptosis with a too high proportional mitochondrial gene expression over the total transcript counts (>2%). Maximal percentage of ribosomal genes was set to 5%. The SCTransform function was used to normalize scale to correct for batch effects with default parameter. The top 15 PCA dimensions were clustered via Seurat’s ‘FindNeighbors’ and ‘FindClusters’ (with parameters: resolution = 0.4). The filtered and normalized matrix was used for the subsequent analysis. Variations in single-cell expression of morphology-associated genes were shown as violin blots.

ciPodocytes were further investigated to identify correlated gene expression. The gene expression correlation was performed with RNA counts via Spearman correlation analysis with the ‘cor’ function from corrplot (v0.92). We clustered the correlation results of podocytes and morphology markers with the ward.D2 method, and visualized them with the corrplot function.

### 2.3. Fixation of Cells for Workflow Imaging

Cells were either fixed with 4% commercially available paraformaldehyd (PFA) solution (Histofix^®^, Roth) for 10 min at room temperature after washing with prewarmed 1× PBS, or fixed with a glutaraldehyde (GA)/PFA solution. For fixation with GA/PFA solution, fixative solution containing PFA (3%) (pH 7.4), washing buffer (final concentration: HEPES 2.38%, CaCl_2_ × H_2_O 0.15%, MgCl × 6 H_2_O 0.20%, saccharose 3.06%, pH 7.4) and glutaraldehyde (2%) needs to be prepared fresh at day of fixation. Here, it has to be noted that the used PFA for this solution needs to be prepared from PFA powder that day, and cannot be ready-to-use PFA, such as, for example, histofix^®^.

For preservation, the cell culture plate was placed on ice, and 50% of the medium was replaced with ice-cold GA/PFA fixation solution. After 10 min, the buffer solution was replaced with fresh 1fixation solution, which then remained on the cells for two hours. Subsequently, cells were washed three times with HEPES buffer, and dehydrated using an ascending ethanol series ([30%, 1 h, 4 °C], [50%, 1 h, 4 °C], [70%, O/N, 4 °C], [80%, 1 h, 4 °C], [90%, 30 min, 4 °C], [100%, 4 × 30 min, 4 °C]).

After dehydration, the cells were either dried in a critical point dryer (CPD) or spiked with hexamethyldisilazane (HMDS), which evaporated without residue. CPD (Leica EM CPD 300; Wetzlar, Germany) drying was performed with specific parameters. The speed of CO_2_ inflow was slow. Between CO_2_ inflow and the exchange cycles, a delay time of 120 s was set. 18 exchange cycles were performed with speed 1. After all ethanol was exchanged, the chamber was slowly heated to 38 °C. 

### 2.4. Light Microscopy

An overview image of the entire coverslip was taken with the Axio Imager M1m (Zeiss, Oberkochen, Germany) for every sample. The 10× objective (NA = 0.2) was used for this purpose. Five cells were selected on each coverslip, and their positions were tagged for further imaging analysis with different devices with a nanoGPS *Oxyo*^®^ tag. Additionally, the cells of interest were imaged with higher magnification, using a 20× (NA = 0.22) and a 50× (NA = 0.55) objective.

### 2.5. Raman Spectroscopy

The Raman spectroscopy analyses were performed on the Horiba LabRAM HR Evolution spectrometer (Horiba Scientific, Kyoto, Japan), in a backscattering geometry under ambient conditions at room temperature. A 532 nm laser focused through a 50× objective (numerical aperture (NA) 0.55, Leica) was used for Raman excitation. The LabRAM system was calibrated to zero spectral order of the grating, which corresponds to specular reflection, and to characteristic first-order phonon band of a silicon wafer at 520.7 cm^−1^. It was performed daily before the use of the system, as well as at regular intervals between the analysis. Before mapping the cells, a power dependency experiment was performed to determine the output power with which the best signal could be detected without damaging the cells.

### 2.6. Scanning Electron Microscopy

All electron micrographs for generation of high-resolution images of podocyte processes were taken with the Zeiss Auriga FIB/FE-SEM (AURIGA TM^®^ Crossbeam Workstation, Carl Zeiss NTS GmbH, Oberkochen, Germany) microscope. In standard conditions, the 30 µm aperture was used to focus the beam. In cases of electrical charges on the sample, a smaller aperture was used. Astigmatism compensation was done at regular intervals between the analysis. The detection of the interaction products of the primary electron beam with the specimen was performed with the secondary electron detector. Due to the low acceleration voltage (1 kV), the working distance was set between 1.5 mm and 2.0 mm.

### 2.7. Atomic Force Microscopy

An atomic force microscope NX20 (Park Systems Corporation, Suwon, Republic of Korea) with a linearized Z-scanner with a dynamic range of approximately 8 μm, decoupled from the feedback-controlled lateral XY-translation stage, was used. In order to minimize wear on both the probing tip and the surface, the non-contact mode was used, and the probe oscillated at its resonance frequency of approximately 309 Hertz (Hz). This reduced the time of physical contact between tip and sample significantly, and also reduces any lateral forces during the scan. An area of 70 × 70 μm^2^ at a resolution of 1024 × 1024 pixels was scanned with a scan speed of 0.1 Hz. Regions of special interest of the scanned cell were then scanned again with smaller scan size.

### 2.8. Scanning Ion Conductance Microscopy

The atomic force microscope NX20, (Park Systems Corporation, Suwon, Republic of Korea), was used for scanning ion conductance measurements. A PBS solution of pH 7.4, corresponding to a physiological solution, was used as the electrolyte in most cases. Both the bath electrode and the pipette electrode, which was placed in the micropipette, are composed of a silver wire, which was coated with silver chloride (Ag/AgCl-electrodes). The voltage applied between the two electrodes in the experiment ranged from 50 mV to 300 mV. After immersing the nanopipette into the solution, further steps had to wait until the current flow was constant. To approach the capillary to the sample surface, a set point that represented the minimal current flow was established that was approximately 1% below the original current flow, and the capillary was then Z-shifted until the specified current flow was achieved. The scan field measured 100 × 100 µm^2^ and was recorded with a resolution of 256 × 256 pixels. Approach-retract scanning mode, in which the pipette is moved vertically and repeatedly approaches and retracts from the sample surface, which is useful for non-contact imaging of the cell surface, was used to control nanopipette position during SICM imaging. To obtain an overview, a scan was then performed within the previously specified parameters. Higher resolution region of interests inside the scan field were imaged afterwards.

### 2.9. Relocation of Cells of Interest Using nanoGPS Oxyo^®^ Tag

In order to find the same location of the sample in the respective microscopes, the coordinates of the ROI selected with the optical microscope (Axio Imager M1m, Zeiss, Oberkochen, Germany) were stored using a nanoGPS Oxyo^®^ tag (HORIBA), and could also be found in the Raman, SEM and AFM after alignment of the devices.

### 2.10. Statistics

All data are expressed as mean ± SEM, where SEM stands, here, for standard error of the mean. For comparison of mean values between two groups, an unpaired *t* test was used. ANOVA was applied for comparison of data from more than two groups. Statistical significance was evaluated using GraphPad Prism. The experimental findings were considered statistically significant if *p* < 0.05.

## 3. Results

### 3.1. scRNASeq Reveals Huge Differences in Gene Expression, Important in Cell Morphology between Individual Podocytes

We performed scRNASeq of differentiated human ciPodocytes, grown on a single petri dish under identical culture conditions, and analyzed the expression of marker genes important for podocyte morphology. Especially, actin cytoskeletal-related genes, such as actinin 4, actin-related protein 2/3, microtubule-associated protein 4, and genes important for morphological cell dynamics, showed high variability in expression between individual cells (Figure 1A). Spearman correlation analysis showed heterogeneity in trends of gene expression within single cells, where some morphology and podocyte genes were found to be regulated, and others were not (Figure 1B).

### 3.2. Samples Substrate and Coating for Workflow with Different Microscopic and Spectroscopic Techniques on the Same Cells

First, a method of sample preparation had to be established that was suitable for LM, Raman spectroscopy, SEM and AFM. In order to select the optimal coating of the substrate for all methods that, especially, met the requirements for Raman spectroscopy, the coverslips were coated with 5 nm of chromium as an adhesion layer, and then platinum, with different layer thicknesses. The goal was to find a coating thickness that was transparent enough to see cells under LM, but also shielding enough. As such, no substrate-specific Raman signal was detected and sample damage was prevented during imaging by the right thermal conductivity. Raman signals from uncoated glass slides were clearly different from the spectrum produced using the 25 nm, 50 nm, or 100 nm platinum-coated substrates (Figure 2A). The uncoated glass slide shows distinct peaks with high intensity, which could possibly cover up podocyte signals. Significantly less signals were detected with 25 nm platinum coating, but the background signal was still too high. The thicker the platinum coating was, the better the background signal of the substrate suppressed. Coating of the glass slides with 5 nm chromium and 50 nm platinum was chosen to be the best option for analyzing podocytes, since the background signal was the weakest, and the cells could still be detected using transmission LM for cell culture growth control.

In summary, glass substrates (12 mm diameter), coated with 5 nm chromium and 50 nm platinum, were used for all methods of the workflow. For induction of podocyte processes, slides were additionally coated with laminin 511, before cells were seeded for differentiation.

### 3.3. Optimal Sample Substrate Fixation and Drying for Workflow with Different Microscopic and Spectroscopic Techniques on the Same Cells

Next, an optimal sample substrate fixation and drying for all techniques of the workflow had to be found [18,19,20]. Sample fixation with 4% PFA solution was suboptimal for the preservation of the ultrastructure of podocytes for all techniques used in the manuscript. Therefore, a modified fixation solution, composed of a combination of GA and PFA according to *Karnovsky*, developed especially for electron microscopy, was tested [21], and resulted in preserved ultrastructure of podocytes, including foot processes and filopodia (Figure 2B(a,b) upper row). Cells fixed with GA/PFA had an intact cell membrane, while cells fixed with the commercially available 4% PFA solution showed distinct holes in the cell membrane (Figure 2B(a,b) lower row). For further development of the correlative workflow, GA/PFA-fixation was used. Sample drying by hexamethyldisilazan (HMDS) was tested and compared to the results from the critical point dryer. HMDS reduces the surface tension, thus bypassing the capillary forces, and subsequently evaporates completely [22]. Electron micrographs of the glass substrates, after drying with HMDS or critical point dryer, showed that more residues remained on the substrate after HMDS drying than with the critical point dryer (Figure 2C). Polyethylene terephthalate (PETG) slides, polyvinylchloride (PVC) slides and glass slides were tested as sample substrates in the critical point dryer. PETG and PVC slides showed clear deformations after the drying process. The size and shape of a glass coverslip, on the other hand, stayed stable during the drying process (Figure 2D). Therefore, glass substrates and critical point drying were used for further development of the correlative workflow. Another challenge was to find the ROIs selected in the LM also in other microscopes, as the fields of view were clearly different. Therefore, a suitable sample holder was needed, which could be used for all microscope stages. A holder was customized based on an existing microscopy holder from the Zeiss company. The Oxyo^®^ tag was firmly attached to the specimen holder, so that the specimen and tag maintained their positions relative to each other. Relocalization accuracy was improved when the tag and ROI were located to each other as close as possible. The stage of each instrument was then calibrated with the different patterns on the nanoGPS Oxyo^®^ tag. For different magnifications of the instruments, different feature sizes were used on that chip. Seven images were taken at random positions on a chosen pattern and fed along with the stage coordinates into the NavYX connect software. The software then automatically converted the sample coordinates and rotation, with respect to the GPS tag (Appendix A). The calibration was performed for all instruments used in the workflow.

### 3.4. Implementation of the Correlative Microscopic Workflow to Characterize Untreated and TGF-β-Stressed Podocytes

Next, we proved our established correlative microscopic workflow on cultured human ciPodocytes left untreated or stimulated with TGF-β. First, an overview image was generated using the LM. The whole sample was scanned in sufficient resolution to resolve the required structures for identification of the cells of interest and selected as ROIs (Figure 3A,B). Even under light microscopy, it was possible to identify long cell processes of untreated ciPodocytes (black arrow in Figure 3A), even though there were minimal limitations in contrast of the sample, most likely due to the substrate coating with chromium, platinum and laminin.

In total, five untreated and five TGF-β-treated podocytes were investigated sequentially in the process, by the use of different microscopic and spectroscopic techniques.

Following ROI determination in the optical microscope, samples were transferred to the Raman spectrometer. After alignment of the nanoGPS Oxyo^®^ tag, the desired ROI could be detected in the field of view. Per cell, 3600 Raman spectra were acquired by mapping, using a step size of 1 µm. A baseline correction was performed to subtract background signal, and the area under the peaks was plotted regarding their intensity. Raman signal intensity was higher in the area of the nucleus, whereas the cytosolic region of the cells provided decreased Raman signal. Raman shifts corresponding to membrane-bound phosphatidylcholine/phosphatidylethanolamine (~722, 760, 766 cm^–1^), phenylalanine (~1002 cm^–1^), phospholipids (~1085 cm^–1^), collagen (~1259 cm^–1^), amid III, cytosin and adenine (~1303 cm^–1^), fatty acids (~1447 cm^–1^) and sphingolipid cluster (~1656 cm^–1^) could be detected in untreated podocytes (Figure 4C). All 3600 Raman spectra per cell were combined into a mean spectrum. The mean spectra of the five cells per group were then combined again. The resulting mean spectrum was then compared to the mean spectrum of the other group. The Raman signal intensity was lower in TGF-β-treated cells than in the non-treated cells, except for two peaks around 813 cm^−1^ and 891 cm^−1^. Spectra of membrane-bound phosphatidylcholine/phosphatidylethanolamine, collagen, amide III/cytosin/adenin, sphingomyelin and fatty acids were significantly lower in TGF-β-treated cells (Figure 4D,E). In addition, some Raman signal peaks were shifted when comparing the spectra of the two groups.

Next, SEM was performed on the same sample sites to analyze ultrastructural differences between exactly the same untreated podocytes and podocytes treated with TGF-β. The same ROIs as in the Raman analysis were relocated with the aid of the nanoGPS Oxyo^®^ tag. [23] Untreated podocytes presented an average of 128 thin cell processes. The mean length of these protrusions was 9.4 µm (Figure 5A–C). In contrast, TGF-β-treated podocytes showed an average number of 89 cell processes. Mean length in the stressed podocytes was 3.88 μm. In summary, the TGF-β-treated podocytes developed, not only fewer, but also shorter, cell protrusions than the untreated control. Moreover, the foot processes of the stressed cells were also thicker than the protrusions of untreated cells (Figure 5A–C).

Afterwards, the ROI of the sample was retrieved in AFM by the nanoGPS Oxyo^®^ tag. (Figure 6A–E). Cell surface roughness calculated by 10% mean-height roughness (Rpv), average roughness (Ra) and difference between the highest “peak” and the deepest “valley” in the surface (Rz) was significantly lower in TGF-β-stimulated podocytes, compared to unstimulated podocytes, indicating a flatter cell surface after stimulation (Figure 6E). Untreated podocytes revealed a median height of about 4 µm, and an average protrusion width of about 200 nm, with this method. After TGF-β stimulation, the median podocyte height was decreased to less than 2 µm, and average protrusion width increased to more than 600 nm (Figure 6A–E).

### 3.5. Sample Preparation for Live-Cell Imaging with a Correlative Microscopic Workflow

Since the dynamics of podocyte processes are of especially great interest, we extended the protocol to a workflow, allowing to characterize the cells alive. This enables, on the one hand, the exclusion of artefacts due to fixation and drying, and, on the other, allows to observe dynamic processes, such as cell-cell communication, -interaction and -contact, and alterations of the same cell to stimulants.

With the help of SICM, it was possible to scan cell surfaces without touching them, and, thus, without the addition of stress to the living cell. Moreover, LM and Raman spectroscopy can be performed on living cells too, so that SICM follow after these methods.

In line with the data from AFM on fixed ciPodocytes, TGF-β-treated cells showed shorter processes that were less often in contact with other cells when measured with SICM when still alive (Figure 7A–C). Protrusions of living ciPodocytes were also significantly wider after stimulation with TGF-β (Figure 7D). Compared to fixed cells, protrusion length and width were both slightly longer/wider in living cells (Figure 5C, Figure 6D and Figure 7C,D).

These results show that TGF-β treatment leads to significant morphological alterations in podocytes, including changes in morphology, retraction of cell processes, loss of cell-cell contacts, and changes in molecular composition, that were all measured in the same cells for the first time.

## 4. Discussion

Podocytes display important functions within the glomerulus that are highly dependent on their sophisticated and specialized morphology. The morphology and surface of podocytes is largely dependent on actin cytoskeletal structures [10]. During development and after injury, foot processes exhibit different actin-based structures, in particular, broad membrane protrusions, called lamellipodia, and long, thin, sharp structures, called filopodia [24,25]. Podocyte injury results in a more motile phenotype in vitro, which is thought to correspond to foot process effacement in vivo [24,26].

Even though ciPodocytes are widely used in the field as in vitro models, they can easily dedifferentiate in culture, and, it has been shown by multiple groups that, ciPodocytes do not express various podocyte-specific markers [27,28]. iPSC-derived podocytes or organoids are recently described alternatives. However, for our purpose, we opted for a cell line that alters morphology after growth factor treatment to have two different conditions to validate our workflow with, and still used ciPodocytes. In the future, our protocol could potentially be adopted to 3D cultures. Especially in organoids, it would be important to analyze the same cells when investigating certain questions with different imaging methods, because of their variable composition and diversity of cell population. Using different samples for multiple methods would harbor the possible error of investigating different things.

Recent rapid development of scRNASeq methods enabled cell type-specific transcriptome profiling. ScRNASeq helps in providing a better characterization of complex tissues, including the heterogeneity within 3D co-cultures. The technique could characterize hidden cell populations, but also revealed high intraindividual differences within a cell population. However, even within podocytes derived from mouse glomeruli, huge heterogeneity was found concerning expression profiles of single podocytes [10]. By scRNASeq, we could show that cell heterogeneity in gene expression also applies for ciPodocytes, even when we only selected cells expressing podocyte markers from the analysis [9,10]. Despite this preselection of cells, we could still detect huge variations in expression levels of morphology-associated genes among individual ciPodocytes that were all cultured in one petri dish under the same conditions, highlighting the need to analyze same samples of cells. Of note, scRNASeq had to be done separately from all subsequent imaging analyses that were performed on exactly the same cells, due to technical reasons, and was not part of the actual workflow protocol.

In morphology of foot process-like structures, lamellipodia and filopodia of podocytes were not much investigated in culture, and the nomenclature is sometimes inconsistent. Even though podocytes in culture usually only express rudimentary foot processes, laminin coating of culture dishes has been shown to promote cell process formation [29,30,31,32]. Therefore, we coated the substrate podocytes that were seeded on with laminin.

Moreover, when podocyte cell surface was studied in the past, it was mostly done with one method only, or, rather, with multiple techniques on different cells. However, as cell-to-cell variability in gene expression and morphology among genetically identical cells are deterministic [9,10,11,12], measuring different cells with various methods harbors the risk of substantial variations. To correlatively link different analytic techniques, one protocol of sample preparation that is suitable for modalities had to be established. We were the first to develop a workflow enabling successively measurements with LM, Raman, SEM and AFM within corresponding cells. In addition to sample preparation, a special sample holder was developed. In order to find the ROIs in different devices, a nanoGPS Oxyo^®^ tag developed by Acher et al. was used [33]. The technology allowed accurate and precise retrieval of the selected point on the sample, with precise calibration of the stage. Furthermore, this technique enables tracking of the same cells in different labs, implying that one lab should not have all these instruments.

First, a method of sample preparation had to be established that was suitable for all measurements, since the different methods have distinct requirements. In the workflow, a reflected LM was used as a first step, which does not demand for special sample preparation. Raman spectroscopy was used as the second technique. Insulating substrates, such as glass slides, result in the inability to dissipate the thermal energy transferred to the sample by the Raman excitation laser beam, resulting in damage to the sample. A heat dissipating, conductive substrate, such as a metal surface, reduces this effect. Furthermore, the intensity of molecular peaks from biological samples is very low in the Raman spectrum. Typical substrates, such as glass or various types of plastic, present characteristic peaks on their own, with higher intensity leading to potential masking of the signal of podocyte-specific molecular vibrations. An already established solution to suppress the substrate background signal is to coat with platinum [34], which reflects the excitation laser light and, thus, prevents the detection of substrate molecular vibrations and sample damage. Similarly, fixation and drying reagents used can alter the Raman signal, as the molecular structure is changed during fixation. Most substrates known to be used in cell culture are insulators. Platinum coating would also serve the purpose of making the substrate conductive for SEM imaging, which is the third microscopic technique of the process. AFM and SICM, which are applied at the end of the workflow, do not require special coating of the substrate, or, rather, sample preparation. However, it has been shown that platinum coating also worked for these techniques [35] and was chosen for our workflow protocol. PFA is most often used for fixation for different microscopy techniques [17,18,19]. However, a combination of GA and PFA preserved cell integrity better in our hands.

When measuring the same cell in different modalities, true cross-correlated statements and a correlation between chemical composition, assessed by Raman spectroscopy, ultrastructural visualization by SEM, and topography calculated by AFM/SICM of individual podocytes was possible. All investigations were carried out with untreated TGF-β-stressed ciPodocytes.

Moreover, we were the first to perform Raman spectroscopy on ciPodocytes [36]. This spectroscopy method is a technique for optical and chemical characterization of the compositional properties of materials. It detects the inelastic scattering of light from molecules, which results in a change in wavelength that corresponds to specific molecular vibrational modes. The platinum coatings of the glass cover slides allowed a better contrast for SEM imaging and, likewise, an improved conductivity of the sample, resulting in decreased background signal. For more stable platinum coating, an interface layer of chromium was used, as, without chromium, the platinum delaminated from the glass substrate in the vacuum of the SEM.

We could assign Raman peaks to membrane-bound phosphatidylcholine/phosphatidyl-ethanolamine, phenylalanine, phospholipids, collagen, amid III, cytosin and adenine, fatty acids and to sphingolipid cluster of untreated podocytes. TGF-β-treated podocytes showed lower Raman signal intensity, shift in Raman signals, and loss of some Raman signal spikes. The lower Raman signal intensity was most likely due to the decrease in podocyte height and volume, detected by AFM and SICM after TGF-β treatment. Shifts and loss of Raman signal are most likely correlated with changes in molecular bindings and composition resulting from altered microRNA, mRNA and protein expression after TGF-β-induced cell stress [37,38,39,40,41,42].

SEM of differentiated cultured human podocytes revealed different cell protrusions branching from the podocyte cell body. These processes were extremely thin and some derived from rudimentary foot processes. Moreover, these processes were in contact with other processes from neighboring cells. It has been reported before that, in health and disease, microvilli protrude from the apical surface of the podocytes into the urinary space [43,44].

Cell protrusions of ciPodocytes were quite long. It has been shown before in podocytes, including ciPodocytes, that laminin accelerates podocyte process formations that were often mutually interlocked [29,30,31,32]. Promotion of filopodia formation, due to laminin coating, has been described in other cells before [45]. Normal width of a foot process, however, is 250–700 nm [46,47]. Filopodia are long, thin and straight cell processes with a diameter of 100–400 nm and a length of 5 to 35 µm [48,49]. Cell protrusions measured in our analysis had a diameter around 200 nm and a mean length of 9.4 µm. In comparison with untreated cells, TGF-β-treated podocytes showed less protrusions that were much shorter in length and lumpier in shape (width of more than 600 nm). As TGF-β was shown to promoted filopodia formation [50], but leads to podocyte effacement, most processes of our cells were most likely foot processes. Thin processes bringing from one cell to another, however, were most likely filopodia.

Next, we were interested in the effect of TGF-β on 3D topography of our ciPodocytes. AFM and SICM are excellent techniques for imaging the topography of living cells with high resolution [51,52]. However, due to unexpected physical contact of the AFM tip with the sample surface, small and fragile structures can be dragged or scratched [53]. In contrast, fundamental operation of SICM relies on an ion current that flows between a pipette electrode and a bath electrode. This ion current is used as feedback signal to maintain a constant distance between a nanopipette and the sample, allowing the pipette to scan the surface for topography imaging without contact [54]. The methodology of measuring podocytes in liquid was established, and the relocalization of the same sample site was also possible here. For this purpose, the development of an incubation chamber for the microscope with 37 °C and 5% CO_2_ was performed. Sample preparation established for our workflow on fixed cells could also be used for the SICM application, leaving out fixation and sample drying. Subsequent fixation and dehydration would still allow for SEM imaging and cross sectioning.

AFM and SICM showed that cell processes were in contact with neighboring cells, forming resembling tight junctions. In line with results seen in SEM analysis, TGF-β-treated podocytes revealed fewer and shorter, but wider, cell processes, and cells were topographically smaller and flatter than the untreated podocytes. Interestingly, cell surface roughness of podocytes was much higher than that reported of other cell types [55], and decreased after TGF-β treatment.

Increasing the sample number might be possible by training AI networks, which can then automatically detect ROIs on a sample in the future. Thereby, the workflow time could be significantly shortened, as selecting suitable cells manually is very time-consuming. This automatization process would drastically increase the reliability of the ex vivo workflow, by allowing many cells to be imaged and analyzed with different methods. After external validation, the protocol might also be of value to investigate iPSC-derived podocytes, 3D models and organoids. Especially in organoids, it would be important to analyze the same cells when investigating certain questions with different imaging methods, because of their variable composition and diversity of cell population. Using different samples for multiple methods would harbor the possible error of investigating different things.

## 5. Conclusions

We showed, with scRNASeq analysis, that ciPodocytes display high variability in expression of morphology-related marker genes, highlighting the need for novel techniques and workflows, allowing the study of cell morphology in one and the same cell with different methods. We established a protocol for the sequential analysis of individual podocyte morphology with LM, Raman spectroscopy, SEM and AFM/SICM, and with the help of nanoGPS tracking (Figure 8). Cellular components/composition could be visualized on cells via Raman spectroscopy and linked to their morphology, with the help of the other microscopes within the same cells. Even though external validation is pending, this protocol allowed intraindividual cell characterization with different high-resolution microscopic and spectroscopic techniques, including methods that have never been used in cultured podocytes before. The use of the nanoGPS Oxyo^®^ technology allows tracking of the same cells in different labs, implying that one lab should not have all these instruments.

## Figures and Tables

**Figure 1 cells-12-01245-f001:**
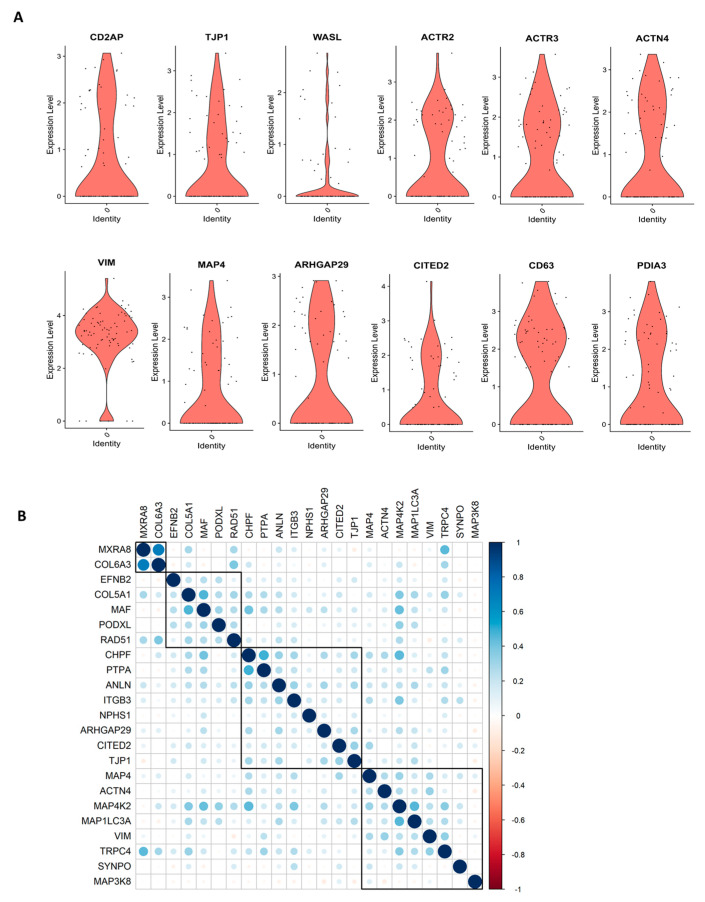
*scRNASeq reveals individual differences in morphology marker of cultured ciPodocytes.* (**A**): Violin plots of expression levels of genes involved in cell morphology and actin cytoskeleton. ACTN: Actinin 4, ACTR3: Actin-related protein 2, ACTR2: Actin-related protein 2, ARHGAP29: Rho GTPase-activating protein 29, CD2AP: CD2 associated protein, ciPodocytes: Conditional immortalized podocytes, CITED2: Cbp/p300-interacting transactivator 2, MAP4: Microtubule-associated protein 4, PDIA3: Protein disulfide-isomerase A3 endoplasmatic reticulum, scRNASeq: Single-cell RNA sequencing, TJP1: Tight junction protein-1, VIM: Vimentin, WASL: Neural Wiskott-Aldrich syndrome protein. (**B**)*:* Spearman’s correlation analysis of gene expression of podocytes and morphology marker genes. Heat map visualization shows the Spearman correlation analysis for podocytes and morphology markers clustered with the ward.D2 method. A strong correlation is blue, and a low correlation is red. ANLN: Anillin, ARHGAP29: Rho GTPase-activating protein 29, *CHPF*: Chondroitin polymerizing factor, PTPA: Protein phosphatase 2 phosphatase activator, *CITED2*: Cbp/P300 interacting transactivator with Glu/Asp rich carboxy-terminal domain 2, COL5A1: Collagen 5A1, COL6A3: Collagen 6A3, EFNB2: Ephrin B2, ITGB3: Integrin beta 3, MAF: Transcription factor Maf, MAP3K8: Mitogen-activated protein kinase kinase kinase 8, MAP4K2: Mitogen-activated protein kinase kinase kinase kinase 2, *MAP1LC3A*: Microtubule associated protein 1 light chain 3 alpha, MXRA8: Matrix remodeling-associated 8, NPHS1: Nephrin, *PODXL*: Podocalyxin like, SYNPO: Synaptopodin, TJP1: Tight junction protein-1, TRPC4: Transient receptor potential channel 4, VIM: Vimentin.

**Figure 2 cells-12-01245-f002:**
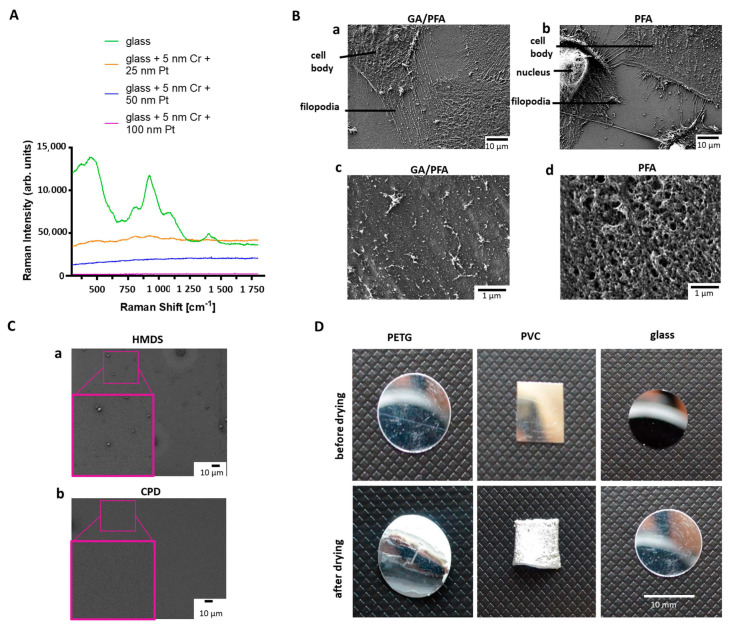
*Comparison of different fixation methods, drying substrates, and coatings for Raman spectroscopy*. (**A**): Raman spectra of the differently coated glass cover slides. Green: uncoated glass; orange: glass coated with 5 nm chromium and 25 nm platinum; blue: glass coated with 5 nm chromium and 50 nm platinum; Purple: glass coated with 5 nm chromium and 100 nm platinum. (**B**): Effects of fixatives on podocyte cell surface and protrusions. Upper row: Effects of fixatives on podocyte cell surface (**B**(**a**)): SEM micrograph reveals intact cell membrane of differentiated cultured human podocytes fixed with GA/PFA solution; Scale bar: 10 µm. (**B**(**b**)): SEM micrograph exhibits perforated cell membrane of differentiated cultured human podocytes, due to fixation with 4% PFA solution; Scale bar: 10 μm. Lower row: Effects of fixatives on podocyte protrusions. (**B**(**c**)): SEM micrograph shows intact ultrastructure of primary and secondary foot processes of samples fixed with GA/PFA; Scale bar: 1 μm. (**B**(**d**)): SEM micrograph demonstrates shorter filopodia of differentiated cultured human podocytes fixed with 4% PFA solution; Scale bar: 1 μm. (**C**): SEM micrographs showing effects of drying. (**C**(**a**)): Glass slide dried with HMDS demonstrates remaining dirt particles; (**C**(**b**)): Glass slide dried in a CPD shows much lower number of particles; Scale bar: 10 μm. (**D**): Images of substrates before (top row) and after drying in CPD (bottom row). Left: PETG slide; Middle: PVC slide; Right: Glass slide. Only glass keeps shape and size during the drying process without deformations; Scale bar: 10 mm. CPD: Critical point dryer, Cr: Chromium, HMDS: Hexamethyldisilazan, GA: Glutaraldehyde, PETG: polyethylene terephthalate, Pl: Platinum, PTA: Paraformaldehyde, PVC: Polyvinylchloride, SEM: Scanning electron microscopy.

**Figure 3 cells-12-01245-f003:**
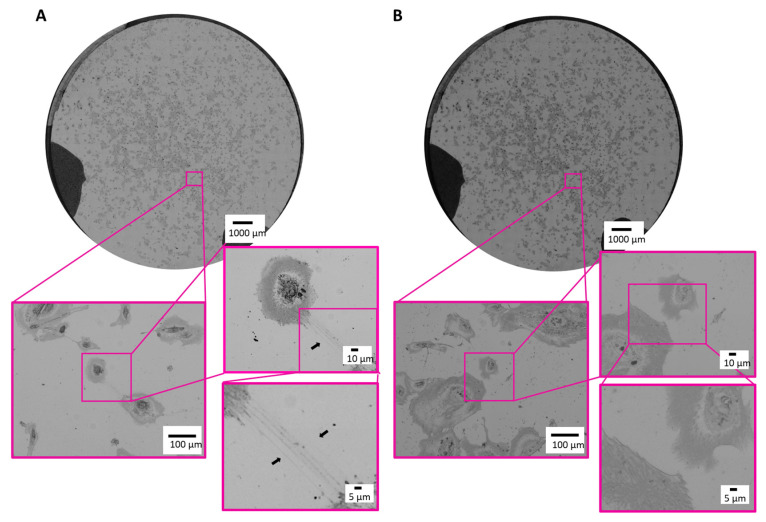
Light microscopic images of the ROI of untreated and TGF-β treated cultured human podocytes. (**A**): Bright field images of the ROI of untreated ciPodocytes in three different magnifications. Black arrow heads point at long cell processes of ciPodocytes reaching a neighboring cell. (**B**): Bright field images of the ROI of TGF-β-treated ciPodocytes in three different magnifications. Top: Overview scan of the whole specimen, scale bar: 1000 μm; bottom: Magnification of cells, scale bar: 100 μm and 10 µm. ciPodocytes: Conational immortalized podocytes; ROI: Region of interest.

**Figure 4 cells-12-01245-f004:**
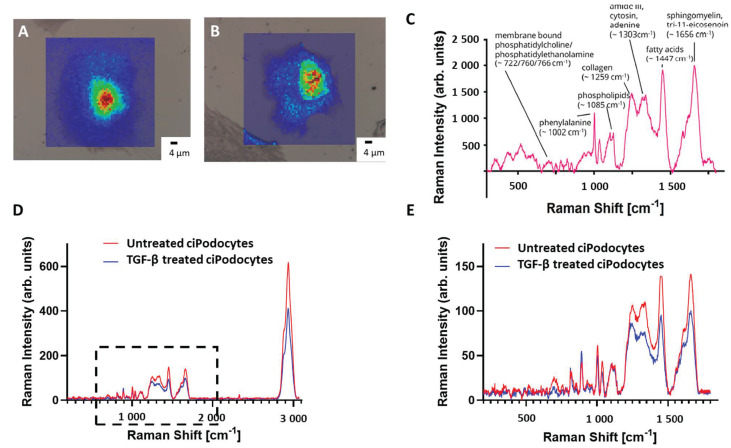
*Raman spectroscopy images of untreated and TGF*-β *treated cultured human podocytes*. (**A**): Raman image of ROI of untreated ciPodocytes. (**B**): Raman image of ROI of TGF-β-treated ciPodocytes. (**A**): Corresponding bright field illumination with heat map of Raman signal intensity of the untreated ciPodocytes. Spectra were recorded at 3600 points, scale bar: 4 μm; Cell nuclei are “lit up”. This is due to more biological material in the nucleus, giving rise to higher intensity Raman signal. (**B**): Corresponding bright field illumination with heat map of Raman signal intensity of the TGF-β-treated podocyte cell. Spectra were recorded at 3600 points, scale bar: 4 μm. (**C**): Mean spectrum of 3600 point-spectra of untreated ciPodocytes, with assignment of the respective molecules. (**D**,**E**): Comparison of whole Raman spectra (**D**) and focus on <1800 cm^−1^ Raman spectra (**E**) of untreated (red) and TGF-β-treated (blue) ciPodocyte. Lower Raman peaks were detected in spectra corresponding to phosphatidylcholine/phosphatidylethanolamine, collagen, amide III/cytosin/adenin, sphingomyelin and fatty acids in the TGF-β-treated ciPodocytes. ciPodocytes: Conditional immortalized podocytes, ROI: Region of interest.

**Figure 5 cells-12-01245-f005:**
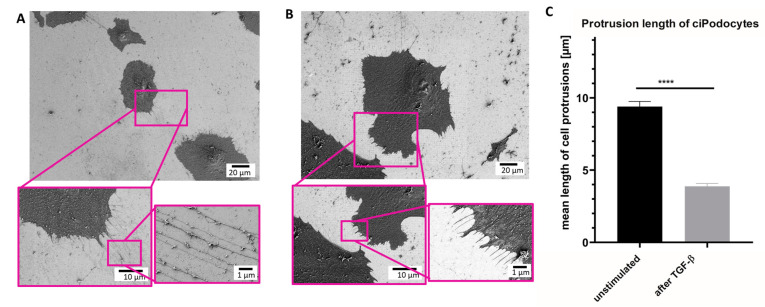
Scanning electron micrograph of ROI of untreated and TGF-ß podocytes cultured human. (**A**): SEM of ROI of untreated ciPodocytes in different magnifications. Top: scale bar: 20 µm; bottom left: scale bar: 10 µm, bottom right: scale bar: 1 µm. (**B**): SEM of ROI of TGF-β-treated ciPodocytes in different magnifications. Top: scale bar: 20 µm; bottom left: scale bar: 10 µm, bottom right: scale bar: 1 µm. (**C**): Quantification of protrusion length in untreated and TGF-β-treated podocytes. **** *p* < 0.0001. n = 638 protrusions in untreated ciPodocytes; n = 445 protrusions in TGF-β-treated ciPodocytes. ciPodocytes: Conational immortalized podocytes, ROI: Region of interest, SEM: Surface electron microscopy.

**Figure 6 cells-12-01245-f006:**
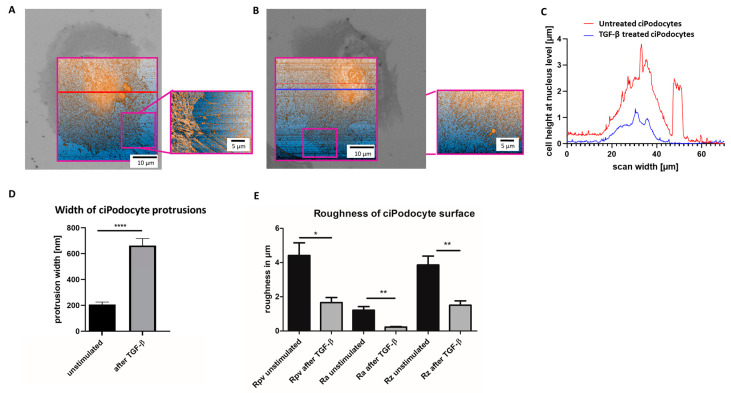
Atomic force microscope scan of the ROI of untreated and TGF-ß treated cultured human podocytes. (**A**): AFM of ROI of untreated ciPodocytes. Left: 70 × 70 μm^2^ scan of the untreated podocyte cell, scale bar: 10 μm. The bright field image from the LM was aligned to the AFM image for clearer assignment of the ROI. Right: Detail scan of filopodia (20 × 27 μm^2^), scale bar: 5 μm. (**B**): AFM of ROI of TGF-β treated ciPodocytes. Left: 70 × 70 μm^2^ scan of the untreated podocyte cell, scale bar: 10 μm. The bright field image from the LM was aligned to the AFM image for clearer assignment of the ROI. Right: Detail scan of filopodia (20 × 27 μm^2^), scale bar: 5 μm. (**C**): Cell height at nucleus level in µm. Red line: Untreated ciPodocytes; Blue line: TGF-β-treated ciPodocytes. (**D**): Quantification of podocyte protrusion width of untreated (black) and TGF-β-treated (gray) ciPodocytes. **** *p* < 0.0001. n = 183 protrusions in untreated ciPodocytes; n = 131 protrusions in TGF-β-treated ciPodocytes. (**E**): Quantification of cell surface roughness of untreated (black) and TGF-β-treated (gray) ciPodocytes. * *p* < 0.05; ** *p* < 0.01. Rpv: 10% mean-height roughness; Ra: average roughness; Rz: difference between the tallest “peak” and the deepest “valley” in the surface. AFM: Atomic force microscopy, ciPodocytes: Conditional immortalized podocytes, LM: Light microscopy, ROI: Region of interest.

**Figure 7 cells-12-01245-f007:**
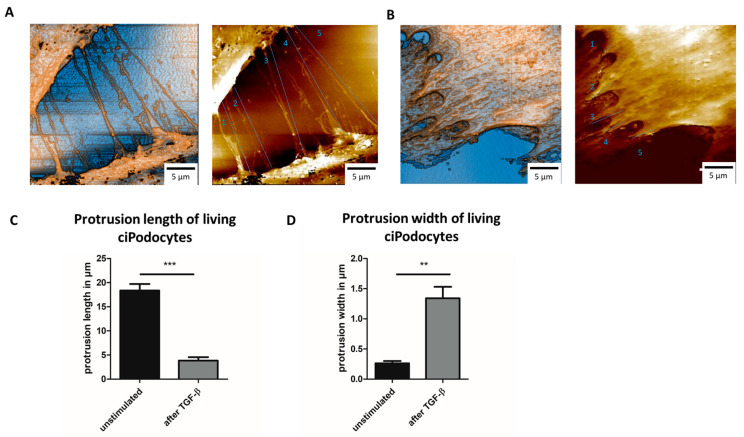
SICM scans of untreated and TGF-β-treated living cultured human podocytes. (**A**): SICM of ROI of untreated ciPodocytes, scale bar: 5 μm. (**B**): SICM of ROI of TGF-β-treated ciPodocytes, scale bar: 5 μm. (**C**): Quantification of protrusion length of untreated and TGF-β-treated living ciPodocytes. *** *p* < 0.001. (**D**): Quantification of protrusion width of untreated and TGF-β-treated living ciPodocytes. ** *p* < 0.01. ciPodocytes: Conditional immortalized podocytes, ROI: Region of interest, SICM: Scanning ion-conductance microscopy.

**Figure 8 cells-12-01245-f008:**
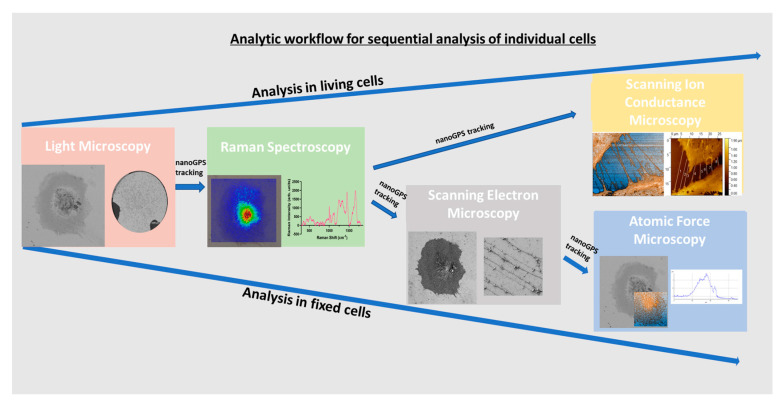
Sequential workflow for analyzing living or fixed cells by different microscopic and spectroscopic techniques within the same cell via nanoGPS tracking.

## Data Availability

The data presented in this study are available on request from the corresponding author.

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
