# Peer review of "Characterizing Intraindividual Podocyte Morphology In Vitro with Different Innovative Microscopic and Spectroscopic Techniques"

_cells, 2023, doi:10.3390/cells12091245_

Round 1

Reviewer 1 Report

The Authors aim to develop a correlative microscopy workflow allowing to measure the exact same cells with different methods, starting from the preanalytical conditions, and defining the best sequence for the different measurements in order to not interfere with the next technique. It combines LM, Raman spectroscopy, SEM and AFM. The protocol was also compatible with measuring living cells.

In the "Materials and Methods" section, the description of the various steps is accurate and exhaustive. The results are clearly described and the discussion is coherent and appropriate. 

Perhaps, the conclusion should be less enthusiastic and the Authors should consider that, as any procedure, also this one needs external validation.

Author Response

We thank reviewer 1 for his/her positive comments on our manuscript. As suggested we revised the conclusions to be less enthusiastic in this point. Furthermore, we included that external validation is needed and hope that other researchers will try our work flow protocol in the future.

Reviewer 2 Report

The authors used conditionally immortalized podocytes (plz mention the source/supplier) to investigate with a consecutive series of analysis methods. By itself this presentation of a 'novel' workflow is interesting although I wonder just how many labs have all these machines at their disposal. 

The techniques used in the manuscript are impressive and it is interesting to see that the authors developed a certain workflow from RNAseq via live imaging to topography and Raman. However, I miss a rationale or guideline when which technique should be used. Live imaging (with/without coating) and SEM did not differ very much and same for AFM. RNAseq does not belong in the workflow because the ROIs did not allow to link the sequence data to specific cells. In my view the strong points are that cellular components/composition can be visualized on cells and linked to their morphology and more distant to their transcriptome. However, this remains a bit dangling in the air in the manuscript.

What is also missing is a sound translation to in vivo: perhaps the authors should not even try that and instead mention that they opted for a cell line that alters morphology after growth factor treatment. After all: podocytes in the kidney are really really very different from ciPodocytes cultured on the tremendously stiff plastic.

To date, there is a strong tendency to move towards 3D culture systems like organs on a chip: how do the authors to envisage to use their workflow on 3D cultured podocytes / glomeruli? This needs to be added to the discussion

The title raised my interest, yet I had expected an in vivo study. While they state themselves that podocytes processes etc alter in glomeruli and while they studied cultured podocytes on plastic, the title should be adjusted and add 'in vitro'. The authors state that they ran techniques on the same cell while in fact they ran techniques on similarly cultured cells. E.g. scRNAseq was done on detached cells. Obviously, it could not have been done differently, yet we still don't know now, which RNA profile belongs to which surface. Also SEM and conductance microscopy are incompatible methods. 

The authors miss an opportunity: indeed they show a certain degree of heterogeneity in expression of various genes in ciPodocytes. But it would be interesting and relevant to show if expression heterogeneity is maintained across genes or e.g. if gene X is high, then also genes Y and Z will be high. Without a proper reference, heterogeneity might simply reflect the nature of these immortalized cells and not be surprising/astonishing at all.

The results are difficult to follow. This is caused by overly and extensive introductions/rationales to the methods they used. This belongs in an introduction or discussion. Results should show, well..., data and results. Thus, the authors should remove/move all redundant text that does not pertain to immediate results.

The results continue in a difficult to comprehend fashion: if LM is the first step in the procedure then why is this lacking from figure 2? The authors need to show LM micrographs before and after treatment for the subsequent methods else the reader cannot compare. Again: add a line that scRNAseq has to be done separately from all subsequent imaging analyses. Fig. 2B show optimizations, but no quantifications or qualifications with clear criteria. How did they decide what is 'best' or better 'optimal'? Fig. 2D: what is the purpose, there is no information other than 'PETG/PVC/Glass' slide...

Evaluation of work flow: effect of TGFbeta: this growth factor certainly does not damage cells but it induces phenotypic changes such as upregulation of collagens, alphaSMA and other mesenchymal markers. At 5 ng/ml it does not induce damage such as apoptosis: that requires >25 ng/ml. 

Fig. 3: are these micrographs obtained after chromium/platinum coating? The authors need to show 'normal' H&E images for comparison because in all images nuclei appear to be detached from the cytoplasm. Also the cells shows a large morphological heterogeneity: from nearly rounded to stretched (irrespective of TGFbeta treatment). This might actually be undelying the observed heterogeneity in scRNAseq. The overview images (top in Fig. 3) show that the ciPodocytes tend to grow in clusters instead of regular monolayers reaching confluency. Obviously, this is of little relevance for the employed workflow of techniques yet for me as a biologist, it does explain things.

Fig. 4: again the authors miss an opportunity: show an LM micrograph next to the Raman images. It seems to me that particularly the nuclei 'light up' in the Raman spectrum but this is not mentioned.

Fig. 5: again I miss the LM micrograph. I also sincerely doubt if a fancy technique like SEM is required to measure podocyte lengths while these are visible in LM already.

Fig. 6: proper comparison is missing - in my own research we did much AFM on surfaces of PFA-fixed cells with excellent results. I find the AFM images that are presented not very convincing and wonder why the authors did not show a comparison with uncoated cells. It seems a bit of a waste to use AFM only to measure the size of the nucleus (expressed as height): this would be much easier: simply measure the diameter and you have the height too (nuclei are generally round balls...). Fig 6F: the authors really need to show convincing pictures of the topography of single cultured podocytes. It seems that the data of Fig. 6D are duplicated in fig. 6F. Therefore 6D should be removed. In Fig. 6C, what is the immense (red) peak on the right side of the graph?

With regard to drying samples prior to SEM, I advise the authors to try tert-butanol. In our hands this worked better (less morphological artefacts) than HDMS and was much faster than critical point freeze drying. Just an advise.

Live-cell imaging: the authors almost 'bend over backwards' with their SICM procedure. SICM is an interesting method, but why was this necessary? After all, to image living cells we have high resolution inverted microscopes at our disposal (phase contrast, DIC etc) as well as even higher resolution confocal laser microscopes. These can also be equipped such that alignment of ROIs is established. I am critical because all cell culture labs have inverted microscopes while SICM is probably a too highly specialized technique.

Minor: plz write scientific English and avoid superlatives such as 'astonishing', ..., ... etc. E.g. heterogeneity found after scRNAseq of ciPodocytes may not come as a surprise because these cells were not synchronized by serum starvation. Also ciPodocytes are but a poor model for human podocytes in vivo: the authors need to confirm their results in primary podocytes or by antibody staining in human kidney sections. To culture any cell type on stiff tissue culture plastic is unphysiological because in vivo tissue (3D) conditions really differ.

Author Response

Dear reviewer,

please find our point by point answer in the attached document.

Thanks a lot for your help to improve the manuscript!

Janina Müller-Deile
